# First-Principle Investigation of Hypothetical NiF$_4$ Crystal Structures

**Tilen Lindič** *[ID], **Anthony Schulz** and **Beate Paulus**

Institute for Chemistry and Biochemistry, Freie Universität Berlin, Arnimallee 22, 14195 Berlin, Germany
* Correspondence: tilen.lindic@fu-berlin.de

**Abstract:** An important synthetic route for the fluorinated organic compounds is electrochemical fluorination (ECF). This is a process taking place on a nickel anode immersed in anhydrous HF. Even though the mechanism is not fully resolved, it is believed that it involves higher valent nickel fluorides formed on the anode. One such compound could be NiF$_4$. Its synthesis and existence have been reported in the literature. However, its crystal structure has so far remained unknown. In this paper, we present, for the first time, the theoretical study of the possible crystal structure of NiF$_4$. We investigated six crystal structures of known metal tetrafluorides as possible candidates for NiF$_4$ by periodic DFT, with the PBE+U method. Of the investigated structures, the most stable polymorph of NiF$_4$ was found to be of the same crystal structure as RuF$_4$. The unit cell parameters were calculated to be a = 4.80 Å, b = 5.14 Å, c = 5.18 Å and $\beta$ = 105.26°. All but one of the investigated structures feature octahedrally coordinated nickel centers with two non-bridging fluorine atoms. In the structure originating from ZrF$_4$, all six fluorine atoms around the nickel centers are bridging and two are located in the vacancies around the nickel skeleton, not directly bound to nickel. The overall magnetic arrangement in all the investigated structures is antiferromagnetic. A comparison with other binary nickel fluorides supports the experimental findings that NiF$_4$ is thermodynamically the least stable.

**Keywords:** NiF$_4$; periodic DFT; investigating unknown crystal structure

## 1. Introduction

Fluorinated organic compounds are, knowingly or not, our indispensable companion in everyday life, be it as surfactants [1], in medicine [2], or as pharmaceuticals [3]. Synthetic routes toward fluorinated organic compounds have therefore been attracting a lot of attention in different branches of chemistry. An industrially well-established method is electrochemical fluorination on a nickel anode immersed in anhydrous HF, first reported by Simons in 1949 [4] and patented shortly thereafter [5]. One of the biggest advantages of the Simons process is its high functional group tolerance. However, despite the success and the vast amount of research on it, the exact mechanism is unknown to this day [6,7]. A proposed mechanism consists of two steps: (1) formation of a Ni$_x$F$_y$ film on a nickel anode under an external potential of around 5−6 V, (2) fluorination of organic substrate by high valent nickel species formed in the previous step [6]. This has been one of the main motivations behind the research on high valent nickel fluorides. The most stable nickel fluoride is NiF$_2$ with a well-described experimental [8] and calculated crystal structure [9]. The next oxidation state, +3 in NiF$_3$, is likewise reported in the literature and the crystal structure is described both experimentally and theoretically [10,11]. The synthesis of two further binary nickel fluorides has been reported in the literature, namely Ni$_2$F$_5$ [12] and NiF$_4$ [10]. Experimental crystal structures of the latter two are unknown. However, we recently published a theoretical investigation of the crystal structure of Ni$_2$F$_5$ [13] and proposed that it adopts the same crystal structure as Cr$_2$F$_5$. A theoretical investigation of

$NiF_4$ has, to our knowledge, not been reported in the literature and is reported in this paper for the first time.

The synthesis of $NiF_4$ was reported by Bartlett et al. [10]. It was synthesized by mixing $BF_3$ in the anhydrous HF solution of $K_2NiF_6$ at $-65\,°C$ which yielded a tan solid precipitate. Above $-55\,°C$ a rapid loss of $F_2$ and formation of $NiF_3$ was observed, which implies that it can be a powerful oxidizer and a source of fluorine for fluorination reactions. Molecular $NiF_4$ was reported recently in joint matrix isolation and computational study [14]. To the best of our knowledge, no other experimental data on $NiF_4$ were published in the literature. Here we present the results on the investigation of a hypothetical crystal structure of $NiF_4$. We have chosen some known crystal structures of $MF_4$ as candidates, where M is the metal atom. Many such crystal structures are known [15] and for our investigation we have chosen those which have a similar ionic radius of the metal atom to that of nickel: $PdF_4$ [16], $SnF_4$ [17], $α$-$ZrF_4$ [18], $OsF_4$ [15], $RuF_4$ [19] and $CrF_4$ [20].

## 2. Computational Details

All the structures were investigated by means of unrestricted, periodic DFT calculations as implemented in the Vienna ab initio Simulation Package (VASP), version 5.4.4 [21–23]. A plane wave basis set with the projector augmented potentials (PAW) was employed [24,25]. 2s, 2p orbitals of fluorine and 3d, 4s orbitals of nickel were treated explicitly with others included implicitly in the core. As an approximation to the exchange-correlation functional, the Perdew-Burke-Ernzerhof (PBE) [26] approach was used because it was shown previously that the bond lengths and crystal structure parameters of $NiF_2$ calculated with this functional are very close to the experimental values [9]. To account for strongly correlated d electrons, a Hubbard correction in the framework of the Dudarev approach [27] was added on top of the PBE functional. The Hubbard U value for $Ni^{4+}$ has previously not been reported in the literature. The value was determined by comparing calculated $Ni-F$ distances with varying U parameter in $K_2NiF_6$ with the experimental values (for a detailed explanation see Supporting information). This way it was established that the optimal U value for $Ni^{4+}$ is 7.7 eV.

To sample the first Brillouin zone, an automatically generated, gamma centerd Monkhorst-Pack K-point grid was used. Gaussian smearing was used for structural relaxation. For the calculation of density of states and the single point energy calculations the tetrahedron method was employed. The plane wave kinetic cut-off energy, K-point mesh, and the Gaussian smearing width for all the investigated structures are shown in Table 1. Convergence tests were performed for all the three values until the total energies were converged to 1 meV.

**Table 1.** Plane wave kinetic cut-off energy (in eV), K-point mesh and Gaussian smearing width (in eV) employed for all the considered parent structures.

| Parent Structure | Cut-Off Energy | K-Point Mesh | Smearing Width |
|:---:|:---:|:---:|:---:|
| $PdF_4$ | 750 | $6 \times 6 \times 6$ | 0.01 |
| $SnF_4$ | 850 | $12 \times 12 \times 6$ | 0.01 |
| $ZrF_4$ | 850 | $4 \times 4 \times 4$ | 0.01 |
| $OsF_4$ | 850 | $4 \times 4 \times 8$ | 0.03 |
| $RuF_4$ | 750 | $4 \times 4 \times 4$ | 0.05 |
| $CrF_4$ | 850 | $6 \times 6 \times 12$ | 0.02 |

The blocked Davidson algorithm was used for the electronic minimization, with the convergence criterion of $10^{-6}$ eV. The conjugate gradient algorithm was used for the ionic relaxation, with the convergence being reached when the forces on individual atoms were smaller than 0.01 eV/Å. Cell shape, ionic positions and cell volume were allowed to relax in the ionic relaxation. To account for possible convergence into the nearest local minimum, starting volumes were set to 80, 100 and 120% of the parent structures. All magnetic couplings within each structure were investigated to determine the most stable magnetic

phase, with starting magnetic moments set to three increasing values to account for three possible spin arrangements of the $d^6$ electron configuration of the Ni cation in the formal oxidation state +4.

To compare the thermodynamic stability of different nickel fluorides ($NiF_2$, $NiF_3$, $Ni^{II}[Ni^{IV}F_6]$, $Ni_2F_5$ and $NiF_4$), the energies per formula unit were normalized to the energy of $NiF_2$ by subtracting the energy of appropriate number of $F_2$. For $NiF_3$ and $Ni^{II}[Ni^{IV}F_6]$: $NiF_3 - 1/2F_2$; for $Ni_2F_5$: $(Ni_2F_5 - 1/2F_2)/2$; for $NiF_4$: $NiF_4 - F_2$. The energy of $NiF_2$ was then set to 0. Because the energies calculated with different Hubbard U parameters are not directly comparable, the average value of U (6.7 eV) for $Ni^{2+}$, $Ni^{3+}$ and $Ni^{4+}$ was used for these calculations. An external code developed by Henkelman et al. [28] was used to calculate the Bader charges. All the structures were visualized by VESTA [29].

## 3. Results and Discussion

The results on the energetic stability of the investigated structures are shown in Table 2. The energy of the most stable polymorph was set to 0 and all the other energies were calculated relative to it. The most stable of the investigated crystal structures is the one derived from $RuF_4$ (I, space group $P2_1/c$). The next two structures, derived from $PdF_4$ (II, space group $Fdd2$) and $OsF_4$ (III, space group $Fdd2$), respectively, are both in the similar energy range, around 0.16 eV per formula unit higher than the structure I. Similarly, the following two structures, derived from $CrF_4$ (IV, space group $P4_2/mnm$) and $ZrF_4$ (V, space group $P4_2/mmc$) lie roughly in the same energy range above the most stable structure. The least stable polymorph is the one derived from $SnF_4$ (VI, space group $I4/mmm$), being 0.8 eV higher in energy per formula unit compared to I.

**Table 2.** Structure numbering, parent structure formulas with their coordination number (CN), energies per formula unit of the calculated $NiF_4$ structures relative to the most stable structure (I: $RuF_4$; in eV), densities of the calculated crystals (in a.u.), magnetic ordering within the crystals (AF: antiferromagnetic), average Ni−F bond distances of the first coordination sphere (in Å), labeled as BD(av.) and calculated band gaps, labeled as BG (in eV).

| Structure | Parent Structure (CN) | Energy | $\rho$ | Magnetic Order | BD(av.) | BG |
|---|---|---|---|---|---|---|
| I | $RuF_4$ (6) | 0.000 | 2.18 | AF | 1.876 | 0.98 |
| II | $PdF_4$ (6) | +0.1605 | 2.16 | AF | 1.891 | 0.87 |
| III | $OsF_4$ (6) | +0.1836 | 2.17 | AF | 1.891 | 0.83 |
| IV | $CrF_4$ (6) | +0.4954 | 1.79 | AF | 1.844 | 0.00 |
| V | $ZrF_4$ (8) | +0.5236 | 2.02 | AF | 2.031 | 0.84 |
| VI | $SnF_4$ (6) | +0.7968 | 1.90 | AF | 1.812 | 0.52 |

Stability of different binary nickel fluorides can only be compared with each other by balancing the number of atoms in different structures. Because $NiF_2$ is the most stable of binary nickel fluorides, it follows naturally to normalize other energies with respect to it (see computational details for detailed description). The energies are shown in Table 3. For the $Ni_2F_5$ and $NiF_4$ only the energies of the most stable polymorph are included. The energy of $NiF_4$ is 1 eV higher compared to the energy of $NiF_2$. This is in agreement with the experimental observation that $NiF_4$ is thermodynamically unstable and decomposes into $NiF_3$ [10]. Furthermore, compared to other binary nickel fluorides, $NiF_4$ is significantly less stable.

The density of each considered structure was calculated as a ratio of molecular mass of one formula unit (in atomic units, for $NiF_4$ 134.69 a.u.) and volume of the crystal per formula unit. The most stable structures, I, II, and III, all have comparable densities, slightly decreasing with the increasing energy (see Table 2). This trend is however broken with structure IV, which has the lowest density but is not the highest in energy. Both structures V and VI continue in the expected trend of decreasing density. When compared to other

binary nickel fluorides (see Table 3), it can be observed that $NiF_4$ has the lowest density, while the most stable $NiF_2$ has the highest density.

**Table 3.** Comparison of of different binary nickel fluorides with respect to their formal oxidation state (OS) of nickel, energy (in eV) normalized to the most stable $NiF_2$, their densities per formula unit (in a.u.), and average $Ni-F$ bond lengths (BD(av.), in Å).

|  | $NiF_2$ [9] | $NiF_3$ [11] | $Ni_2F_5$ [13] | $Ni^{II}[Ni^{IV}F_6]$ [30] | $NiF_4$ |
|---|---|---|---|---|---|
| OS (Ni) | +2 | +3 | +2/+3 | +2/+4 | +4 |
| Energy | 0 | +0.2302 | +0.3297 | +0.5674 | +1.0091 |
| $\rho$ | 2.83 | 2.49 | 2.56 | 2.48 | 2.18 |
| BD(av.) | 2.021 | 1.883 | 2.085/1.874 | 1.984/1.829 | 1.876 |

### 3.1. General Comparison among the Different NiF₄ Polymorphs

In all investigated structures, nickel centers are octahedrally coordinated, as in the other known binary nickel fluorides. Three pairs of almost equal $Ni-F$ bond lengths are observed in all the structures. With the exception of structure V, there are two non-bridging fluorine atoms in all the structures. Structure V originates from 8-fold coordinated metal atoms. However, after the structural relaxation all the nickel atoms are octahedrally coordinated with all six fluorine atoms around the nickel center being bridged and the two almost free (non-binding) fluorine atoms sitting inside the crystal not directly coordinated to any Ni centers. Only the most stable structure and structure V will be discussed here in detail (for a brief discussion of the others see the supporting information). The average bond distances within the octahedra in the investigated structures are shown in Table 2. It can be seen that they are roughly the same in all the structures, except for the structure V where the average $Ni-F$ bond length is around 0.15 Å larger. In comparison with other binary nickel fluorides (see Table 3) all the investigated structures, except for V, show shorter average bond lengths compared to formal Ni(II) in $NiF_2$. Interestingly, the average bond lengths in the compounds with nickel in the formal oxidation state +3 ($NiF_3$ and $Ni_2F_5$) are very similar to the average bond lengths in the investigated structures of $NiF_4$.

With respect to the calculated band gaps, all the structures, except for structure IV, are semiconductors (see Table 2). The most stable calculated $NiF_4$ polymorph I has the highest band gap of almost 1 eV. The band gaps of structures II, III, and V are roughly 0.1 eV lower. The least stable polymorph originating from the crystal structure of $SnF_4$ has approximately half smaller band gap of 0.52 eV. Interestingly, structure IV, originating from $CrF_4$ is metallic and does not have a band gap. In comparison to the PBE+U calculated band gaps of $NiF_2$ (band gap of 4.6 eV [9]) and $NiF_3$ (band gap of 2.54 [11]), the calculated band gaps for all the investigated $NiF_4$ structures in this study are significantly lower. However, a trend of decreasing band gap size with increasing formal oxidation number of nickel can be observed. It should be noted that despite the Hubbard correction to the PBE functional, the band gaps calculated with the GGA functionals are usually underestimated compared to the hybrid functionals and even more so compared to the experimentally measured band gaps. However, the PBE+U calculations still reproduce the correct trend, as the predicted small band gap of around 1 eV would correspond to the visible light adsorbing material which would result in a black color, as seen for the active $Ni_xF_y$ film in the Simons cell. Therefore, our calculated band gap additionally supports the presence of Ni(IV) centers.

### 3.2. Structure I

The structure of the most stable investigated $NiF_4$ is shown in Figure 1. It is a monoclinic structure, belonging to the space group $P2_1/c$. Its lattice parameters are a = 4.80 Å, b = 5.14 Å, c = 5.18 Å and $\beta$ = 105.26°. The magnetic ordering within the structure is antiferromagnetic, with the total net magnetic moment being 0. Its structure is layered and can be described as a network of puckered sheets. Each nickel atom is six-fold octahedrally coordinated. Four equatorial fluorine atoms are bridging, while the axial two point in-between

the sheets. The octahedra are tilted in such a way that the fluorine atoms pointing in the space between the sheets are the furthest apart.

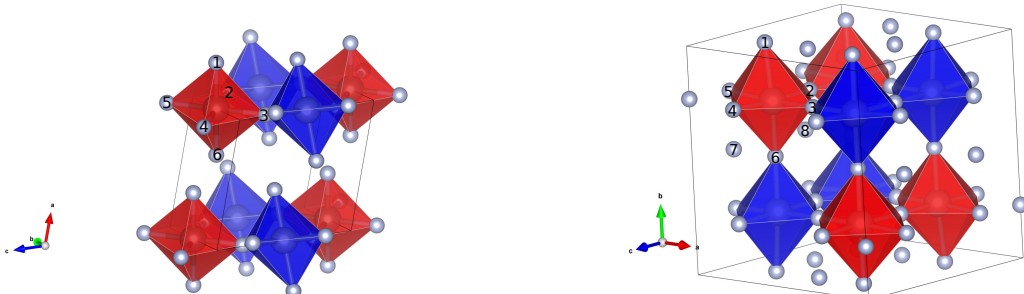

**Figure 1.** Structures of the most stable structure I derived from $RuF_4$ (**left**) and the structure of V (derived from $ZrF_4$ (**right**)), both shown with octahedra. The blue and red colors represent the opposite orientations of magnetic moments on the Ni centers.

Bond distances and bond angles in structure I are shown in Table 4. There are three pairs of equal nickel fluorine bond distances, with two of the pairs being almost equal and one equatorial distance around 0.1 Å larger. The octahedron is very close to having perfect angles, with only two F−Ni−F angles deviating by 5° (for the exact numbers see supporting information). Compared to other binary nickel fluorides the bond lengths are smaller than in $NiF_2$ (see Table 3); however they are roughly the same as in the $NiF_3$. In comparison to the reference compound $K_2NiF_6$, where the Ni−F bond length is 1.777 Å, it can be seen that in $NiF_4$ structure I the bond lengths are significantly elongated. This can be explained with the different spin state of the nickel ion. In $K_2NiF_6$ the d electron configuration is low spin, whereas in $NiF_4$ the electron configuration is high spin, which causes elongation of the Ni−F bond lengths. Another reason may be the fact that even though formally nickel is in oxidation state +4 in $NiF_4$, in the calculated structures the oxidation state lies somewhere between +3 and +4 (see the discussion on magnetism below).

**Table 4.** Nickel fluorine bond distances (in Å) in the octahedra in structure I and V, respectively. Numbering of the fluorine atoms is the same as in Figure 1. For structure I the bridging and non-bridging fluorines are denoted in parentheses as b and n, respectively. In structure V all the fluorine atoms are bridging and the labels are omitted.

| Structure I | | Structure V | |
|---|---|---|---|
| **# F** | **Distance** | **# F** | **Distance** |
| 1 (n) | 1.860 | 1 | 2.107 |
| 2 (b) | 1.933 | 2 | 1.986 |
| 3 (b) | 1.836 | 3 | 1.980 |
| 4 (b) | 1.933 | 4 | 1.988 |
| 5 (b) | 1.836 | 5 | 2.133 |
| 6 (n) | 1.860 | 6 | 1.990 |

The overall magnetic arrangement in I is antiferromagnetic. The magnitude of the magnetic moments on the nickel atoms is 2.36 $\mu_B$. Opposite orientation of magnetic moments are depicted as red and blue octahedra in Figure 1. It can be seen that in the chains of corner sharing octahedra in each of the puckered sheets the arrangement is also antiferromagnetic, with no direct local ferromagnetic interaction. The mechanism of interaction can be described in terms of superexchange through the fluorine atoms. There are two antiferromagnetic superexchange angles, both almost equal at 151°.

The band gap of the of structure I is 0.98 eV, making it a semiconductor. A plot of the orbital projected density of states is shown in Figure 2. It can be seen that there is little hybridization between the fluorine and the nickel states. In the valence region, most of

the p and d orbital states are clearly separated, whereas in the conduction region there is some overlap between both. At the Fermi level fluorine p orbitals are dominating. In the conduction region, the fluorine p band probably corresponds to the non-fully filled orbitals of non-bridging fluorines (see the general discussion on magnetism).

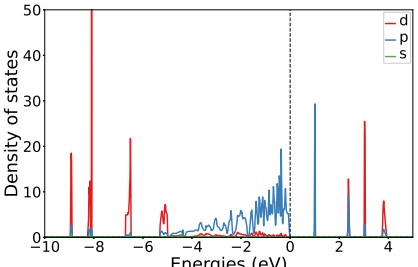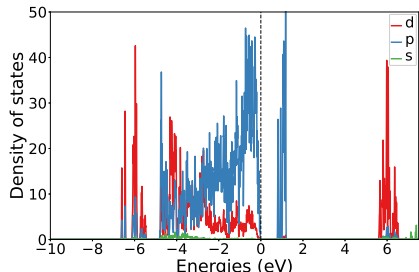

**Figure 2.** Orbital resolved density of states plots for structure I (**left**) and V (**right**). NB: Because of overall antiferromagnetism the plots are symmetric with respect to the x-axis and only the upper part is shown.

### 3.3. Structure V

Structure V, deriving from the structure of $ZrF_4$, original space group $P4_2/m$ is interesting and worth discussing because in its parent form, each zirconium is 8-fold coordinated. By substitution of zirconium with nickel the coordination number decreases to 6, thereby rendering some fluorine atoms not directly coordinated to any of the nickel centers. The space group of the $NiF_4$ structure V after relaxation is $P4_2/mmc$, hence possessing a symmetry compared to the parent structure, but staying tetragonal with all the unit cell angles equal to 90°. The lengths of the unit cell are a = b = 8.20 Å and c = 7.95 Å. In contrast to other structures, all six fluorines surrounding the nickel centers are bridging (see Figure 1). In the conventional unit cell, there are eight formula units. In this cell, there are in total eight fluorine atoms which are not directly connected to any of the nickel centers, one per formula unit. Each octahedron is connected to the neighbors via all of its fluorine atoms, forming a cube-like structure with the nickel atoms sitting in the corners of the cube. The fluorine atoms which are not directly connected to nickel centers sit in the middle of these cubes, in an alternating fashion; that is in every second cube.

Compared to other structures, the local Ni−F distances are much longer (see Table 4) This can be explained by the magnetic structure (see discussion on magnetism below). In terms of angles, the octahedra are slightly distorted, with angles deviating by around 7° (for the values see Supporting information).

The overall magnetic ordering is antiferromagnetic. In contrast to the structure I, where the antiferromagneticity can be observed in the row of connected octahedra, the interaction is more complicated. In the c-direction the interaction is ferromagnetic (see Figure 1) and in the other two directions the coupling is antiferromagnetic. There are two superexchange angles, the ferromagnetic superexchange angle is 175.81° and the antiferromagnetic is 178.63°.

The band gap of structure V is 0.87 eV. Its orbital resolved DOS is shown in Figure 2. Similarly, as for structure I, the band gap arises from fluorine p states. A significant hybridization between the fluorine p and nickel d states can be seen in the valence region. Nickel d states in the valence region are quite delocalized, whereas in the conduction region they are mostly localized around 6 eV. Strongly pronounced p states of fluorine are probably arising from the non-connected fluorine atoms.

### 3.4. Magnetism

In a perfect octahedron (with the perfect local $O_h$ point group symmetry), d-orbitals split into two levels, three $t_{2g}$ which are lower in energy and two $e_g$ levels which are higher in energy. Electron configuration of nickel in formal oxidation state +4 is $d^6$. An orbital diagram of splitting is shown in Figure 3. There are three possible electron configurations.

High spin, with four unpaired electrons (S = 2), "medium" spin configuration with two unpaired electrons (S=1) and low spin configuration, where all the electrons are paired (S = 0). The perfect calculated magnetic moments would therefore be 4, 2, and 0 for each of the possible configurations, respectively. From experience with the nickel fluorides, it can be expected (except for the low spin configuration) to see slightly smaller values than formally assigned (for example for $NiF_2$ with two unpaired electrons the calculated value of magnetic moment is around 1.8 $\mu_B$ [9]).

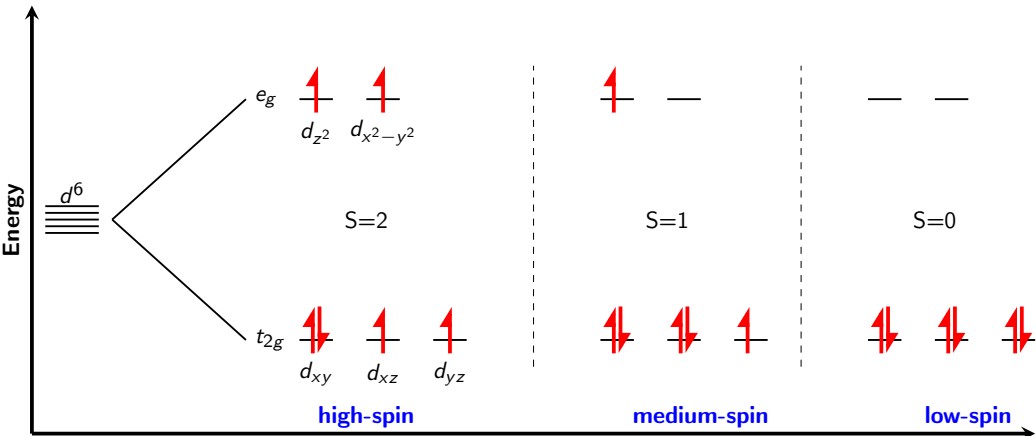

**Figure 3.** Splitting of d orbitals in the perfect octahedral ligand field, with possible electron configurations for $d^6$ $Ni^{4+}$ cation.

The magnitudes of the most common magnetic moments and Bader charges for nickel atoms, bridging and non-bridging fluorine atoms, respectively, are shown in Table 5 (for a complete list of all the atoms in the investigated structures see the supporting information). The magnetic moments on the nickel atoms range from 1.9 to 2.7 $\mu_B$. This points to a high spin electron configuration, however deviates significantly from the formal assignment. A clear difference in the magnitude of magnetic moments between the bridging and non-bridging fluorine atoms is observed. The bridging fluorine atoms have magnetic moments around 0, which indicates that each of this fluorine atoms accepted one electron and can be regarded as closed shell anionic $F^-$. This is also the case in all the other binary nickel fluorides ($NiF_2$, $NiF_3$ and $Ni_2F_5$), where all the fluorine atoms are bridging.

In a simplified picture, the bridging fluorine atoms can be thought of as receiving half an electron from each of the nickel centers they are attached to, which means that four such bridging fluorines would take two electrons from one nickel, leaving a $Ni^{2+}$. To come to nickel in the oxidation state +4, two further electrons have to be removed, which should happen through the remaining two non-bridging fluorine atoms. These would be expected to show magnetic moments around 0. In the investigated structures these are observed to be around 0.5, indicating that between the two non-bridging fluorines slightly more than one electron is shared, and therefore the nickel atom is probably somewhere between the formal oxidation state of +4 and +3, which can explain lower than expected magnetic moments on the nickel atoms. One of the explanations of the low stability of $NiF_4$ and the experimental observation of its rapid transformation into $NiF_3$ and loss of $1/2F_2$ above $-55\,°C$ can be that, even though the formal oxidation state of nickel is +4, such a high oxidation state cannot be achieved. The true oxidation state lies somewhere between +4 and +3 and the formation of energetically more favourable $NiF_3$, with nickel in a pure +3 oxidation state is readily driven by the loss of fluorine.

By following this simplified picture of counting electrons, two possibilities to trap nickel in the pure oxidation state +4 can be thought of. The first one is that nickel is 4-fold coordinated and the square planar units are not connected with each other. This is very unlikely, because such a structure could not lose $1/2F_2$ and transform into octahedrally coordinated $NiF_3$, as observed in experiment. The second possibility would be for the nickel atom to be 8-fold coordinated with all the fluorine atoms being bridging.

As described, structure V originated from initial 8-fold coordination around the metal (vide supra). In the structural relaxation the coordination number around the nickel decreases and two fluorine atoms are not directly coordinated to any of the nickel centers, having one unpaired electron each with the magnetic moment of around 0.8 and Bader charge of 0, clearly indicating a neutral fluorine atom. Formally, the oxidation state of nickel in this case is 3+ which is reflected in lower magnetic moment on nickel compared to the other structures. This leads us to believe that nickel is not large enough to accommodate 8-fold coordination and hence renders also this type of structures not feasible for $NiF_4$.

**Table 5.** The most frequent values for the magnitude of magnetic moments (in $\mu_B$) and Bader charges for all the investigated structures, for nickel atoms, bridging fluorine atoms and non-bridging fluorine atoms. Magnetic moments are shown in the first rows and corresponding Bader charges in the second.

| Structure | Quantity | Ni | F (Non-Bridging) | F (Bridging) |
|---|---|---|---|---|
| I | $\mu$ | 2.4 | 0.6 | 0.0 |
| | $q_B$ | +1.8 | −0.3 | −0.6 |
| II | $\mu$ | 2.3 | 0.6 | 0.0 |
| | $q_B$ | +1.7 | −0.3 | −0.6 |
| III | $\mu$ | 2.4 | 0.6 | 0.2 |
| | $q_B$ | +1.7 | −0.3 | −0.6 |
| IV | $\mu$ | 2.4 | 0.5 | 0.2 |
| | $q_B$ | +1.6 | −0.3 | −0.6 |
| V | $\mu$ | 1.9 | 0.8 & 0.4 | 0.2 |
| | $q_B$ | +1.5 | 0.0 & −0.4 | −0.5 |
| VI | $\mu$ | 2.7 | 0.5 | 0.2 |
| | $q_B$ | +1.9 | −0.3 | −0.6 |

## 4. Conclusions

In this work, we investigated the possible crystal structures of unknown $NiF_4$ by periodic DFT. Six different known structures of $MF_4$ type were considered, namely $RuF_4$, $PdF_4$, $OsF_4$, $CrF_4$, $ZrF_4$, and $SnF_4$. Of the investigated structures, the one originating from the $RuF_4$ was found to be the most stable. The nickel centers are octahedrally coordinated and arranged in a layered structure, which can be described in terms of a network of puckered sheets. Two of the fluorine atoms are non-bridging and four of them are bridging. The most stable magnetic phase was found to be antiferromagnetic, with a high spin electron configuration. Analysis of magnetic moments in all the structures pointed to the fact that even though the formal oxidation state in $NiF_4$ is +4 the calculated oxidation state lies somewhere between +3 and +4, which could be an indication of why the $NiF_4$ is not stable. In this regard, an especially interesting structure is V, originating from a $ZrF_4$ which consists of 8-fold coordinated metal centers and upon the exchange of Zr with Ni becomes 6-fold coordinated with some fluorine atoms not directly connected to the Ni centers.

Comparison with other binary nickel fluorides showed that $NiF_4$ is the least stable, which is in agreement with experimental findings. However, taking into account the nature of the Simons process, which takes place at the applied external potential and was the motivation to study higher valent nickel fluorides, it can be assumed that $NiF_4$ can be locally available in the $Ni_xF_y$ film formed on the nickel anode. This is further corroborated by the studies of $NiF_2$ surfaces with the access of fluorine, which showed that at applied potential the surfaces are further stabilized by the adsorption of fluorine [9]. In this study, we showed that even in the bulk phases of $NiF_4$, such fluorine sources are available within the even slightly less stable polymorph of $NiF_4$.

Unfortunately, no further experimental data are available to verify our model; however, we believe that our model can shed some light onto both experimental and theoretical future investigations of high valent nickel fluorides. With this study we also added the

last piece into the puzzle of theoretical investigations of experimentally known binary nickel fluorides, since $NiF_4$ was, until now, the only one which was not yet investigated theoretically.

**Author Contributions:** Conceptualization, T.L. and B.P.; methodology, T.L. and A.S.; validation, T.L., A.S. and B.P.; formal analysis, T.L.; investigation, T.L. and A.S.; data curation, T.L. and A.S.; writing—original draft preparation, T.L.; writing—review and editing, T.L., B.P and A.S.; visualization, T.L.; supervision, T.L. and B.P.; project administration, B.P.; funding acquisition, B.P. All authors have read and agreed to the published version of the manuscript.

**Funding:** This research was funded by the Deutsche Forschungsgemeinschaft (DFG, German Research Foundation)—Project-ID 387284271—SFB 1349.

**Data Availability Statement:** See supporting information. Further data are available from the authors on a reasonable request.

**Acknowledgments:** The authors would like to thank North German Supercomputing Alliance (HLRN) for the computational time and the German Research Foundations (DFG) for funding within the SFB 1349—fluorine specific interactions.

**Conflicts of Interest:** The authors declare no conflict of interest.

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
