# Peer review of "First-Principle Investigation of Hypothetical NiF4 Crystal Structures"

_crystals, doi:10.3390/cryst12111640_

Round 1
Reviewer 1 Report
In this work, the crystal structure of NiF4, a strongly oxidizing inorganic compound, was predicted computationally by Tilen Lindic et al. The possible polymorphism of NiF4 and the magnetic properties were also predicted by the author. The report of this structure provides an important reference for scientists in electrochemical organic synthesis in describing the structure of intermediates for substrate adsorption on nickel anodes, etc. I think this work deserves to be published in the Crystal journal after minor modifications. Since it is predicting the structure of NiF4, a thermodynamically unstable substance, it should match the information on NiF4 in the experimental literature. As the authors cite in the main text of Ref. 10 (J. Am. Chem. Soc. 1995, 117, 10025-10034.), NiF4 is a tan solid that does not give off F2 at -60°C and decomposes to P-NiF3 at > -55°C, or the information here is supposed to be computationally simulable. 1) The authors have calculated the density of states of monoclinic and tetragonal phases. Please calculate the UV-visible spectra of each phase of NiF4 separately if it is possible. Then predict the color of each phase based on the calculation spectra and compare it with the experimentally reported tan color. 2) Please calculate the thermodynamic data of each phase of NiF4, as well as the thermodynamic data of P-NiF3, and also combine the thermodynamic data on F2 experimentally to calculate the temperature at which the decomposition reaction of NiF4 occurs, and compare it with the experimentally reported -60°C as well as -55°C. In addition, I believe that the following sections also need to be discussed and revised to enhance the scientific value of the manuscript. (3) The general form of octahedral distortion in the D6 configuration should be that the metal-ligand bond in the XY-direction is longer than the metal-ligand bond in the z direction normally. But the octahedra of Structure 1 shown in both Figure 1 and Table 4 are elongated in the z-direction. So here further discussion about the anomaly deserved is needed. 4) The authors describe the splitting of the d orbitals in the octahedral field in Figure 3 and use this to continue the discussion of the magnetic properties of the individual structures. With the current description, only S=0 can be valid in such a schematic according to the energy minimum principle, but the phenomena of S=2 and S=1 can also be valid in the actual system. This is because the octahedron produces distortions and its point group is no longer Oh symmetry. This also means that the relative energy between the orbits and the concurrency of the orbits is no longer the same as the Oh symmetry, so the representation of the orbits is no longer applicable to "eg" and "t2g". The suggestion is to modify Figure 3 by studying the article: Gelessus, A.; Thiel, W.; Weber, W. Multipoles and Symmetry. J. Chem. Educ. 1995, 72 (6), 505. DOI: 10.1021/ed072p505.Author Response
We thank the reviewer for taking the time to carefully read the manuscript. We thank for the positive feedback and the valuable suggestions. We have marked all the proposed changes in the manuscript in red.
1.) Thank you for the suggestion. We have also calculated the DOS of other investigated polymorphs and they are presented in the Supplementary information. The method used was PBE+U which is known to underestimate the band gap (from the experience from our group with NiF2 and NiF3). We have tried to do the hybrid functional calculations (HSE06), but the calculations turned to be computationally too expensive and therefore not feasible. We think that such colour comparison as suggested with the PBE+U method would therefore not give meaningful results.
2.) We thank for the suggestion. The relative stabilities of different binary nickel fluorides are given in Table 3. We call them thermodynamic data, because it is only the electronic energy at T = 0 K. We assume that the temperature would indeed have a major impact on the thermodynamic data, but to calculate this, phononic calculations would be necessary, which we deemed to be beyond the scope of this manuscript.
3.) We thank for this comment. Because we started from known crystal structures of different metal (IV) fluorides, we have used the same labelling of the crystallographic axes as in the parent crystal structures. It was our intend to stay consistent with the parent compounds and not change to the conventional labelling of the axes, where the most deviating Ni-F distance would indeed be in the z-axis direction.
4.) Thank you for the comment. We have amended the text in such a way that it is clear that Figure 3 shows the splitting of d-orbitals in a perfect Oh point group symmetry environment and only shows an idealised picture. However, we think that this idealised picture can aid the discussion of the results.
Reviewer 2 Report
Organofluorine compounds are widely used in modern science and technology. Fluorinated polymers are widely used.
Electrochemical processes for the production of organofluorine are quite high-performance and selective, but sometimes their mechanism is not completely clear. In this regard, the theoretical study of one of the products of the conversion of nickel electrodes is extremely important.
However, the authors report that they did not find information about previous studies on this topic. Alas, even a cursory Google search suggests that this is not the case.
I would encourage the authors to more carefully search the relevant literature and provide comparisons of their data with previous ones. The article will only benefit from this.
I highly recommend you take a look at the following document:
Zemva, B., et al. "Thermodynamically unstable fluorides of nickel: NiF4 and NiF3 syntheses and some properties." Journal of the American Chemical Society 117.40 (1995): 10025-10034.
In general, due to the novelty of the presented results, the manuscript can be accepted for publication after appropriate changes.
Author Response
We thank the reviewer for evaluating the manuscript and are grateful for the valuable suggestions. Regarding the suggested reference we would like to point out that the reference is already included in the manuscript (reference 10) and also discussed in the introduction (this reference is the only reference on the synthesis of NiF4 available in the literature). In the introduction we have added a recent paper which was the first one to report the existence of molecular NiF4 (marked in red).
Reviewer 3 Report
Comments: Crystals-2033271
In the present study, the author presents the solution of the First-Principle Investigation of Hypothetical NiF4 Crystal Structures. The subject of the study is interesting and well-developed. I recommend the publication of the manuscript after minor revision. My suggestions are listed below:
1. Novelty of the problem is missing. Please add in the revised version. What a significant exploration prior to published works.
2. Abstract should be enhanced with some major results.
3. Add some recent papers related to your study. The authors used old papers.
Author Response
We would like to thank the reviewer for evaluating the manuscript and for the overall positive feedback. We have included the suggested changes in the manuscript and marked them in red.
1. We have amended one sentence in the introduction saying: A theoretical investigation of NiF4 has, to our knowledge, not been reported in the literature and is reported in this paper for the first time.
2. Thank you for the suggestion. We have added another sentence about the thermodynamic stability and stressed that we report the crystal structure of NiF4 for the first time.
3. Thank you for the suggestion. Unfortunately we are not aware of any newer publications regarding the synthesis or the crystal structure of NiF4. We have added a reference related to the matrix isolation study of NiF4, which is also the only report on molecular NiF4 in the literature.